



# Building a Diverse and Equitable Distributed Wind Workforce: A Strategic Approach to Collaborator Selection

Kendall Parker[1], Kamila Kazimierczuk[1], Micah Taylor[1], Danielle Preziuso[1], Andrew White[1]

[1]Pacific Northwest National Laboratory, Richland, WA, 99354, USA

*Correspondence to*: Dr. Kendall Parker (kendall.parker@pnnl.gov)





**Abstract.** The demand for a skilled distributed wind (DW) workforce is rising with industry growth and recent federal support for technology adoption. However, challenges persist in scaling the industry. For example, DW installers have reported difficulty hiring, and areas with economically viable DW potential are often in rural and remote disadvantaged
communities where workforce development opportunities have not been fully realized. Overall, the wind energy sector has a below-average representation of marginalized groups, and the transition to a cleaner energy future is an opportunity to change that. As more renewables, including DW, come online, scaling workforce capacity can be done in tandem with supporting workforce diversity. Moreover, to promote fair and equitable outcomes in workforce development, efforts to address limited workforce capacity should encourage participation from under-resourced and under-represented populations.
Engaging under-represented populations not only helps close skills gaps but also ensures that the wind energy sector benefits from diverse perspectives, driving innovation and more effective solutions. Additionally, prioritizing workforce diversity ensures marginalized communities share in the benefits of the clean energy transition, ultimately supporting the long-term sustainability and inclusivity of the industry. The Diverse and Equitable Workforce in Wind Energy (DEWWind) project has developed a replicable equity-driven rubric to identify potential industry and academic collaborators for workforce
development programming. This rubric identifies and considers workforce partners outside of traditional networks across locational, institutional, and socioeconomic criteria to advance new partnership-building opportunities in areas favorable for DW. These collaborative opportunities can serve as case studies for improving future scale-up of equitable wind workforce partnerships.

## 1 Introduction

Wind energy is the largest source of renewable electricity in the United States in terms of cumulative installed capacity and is one of the fastest-growing sources of electricity overall—requiring a skilled workforce to support industry growth (Climate Central, 2024; WindExchange, 2024; ACP, 2024). Technological maturity, improvements in advanced manufacturing, alongside cost reductions making wind cheaper than conventional fossil fuels, have stimulated growth across wind sectors. Policy momentum is also stimulating wind workforce development. The Inflation Reduction Act (IRA)
provides up to 30% credit for eligible investments in wind projects that adhere to prevailing wage standards and employ apprentices from Department of Labor (DOL) registered apprenticeship programs (EERE, 2023; DOL, 2024). Federal decarbonization targets and state renewable portfolio plans have further elevated wind energy as a key part of the larger energy transition.

Deploying wind energy technologies at the distribution level of the grid, commonly called distributed wind (DW) (Preziuso
et al., 2022), has been primed for growth. Unlike land-based and offshore wind, which provide power to distant end-users, DW stays relatively local—built in the communities and backyards of the individuals using its power, with technology sized to the application. DW utilizes small, mid, and large (i.e., utility-scale) turbines to serve onsite power demand or local loads (DOE-WETO, 2020). While utility-scale land-based and offshore wind represents the largest chunk of installed generation





capacity, DW is a growing part of this wind energy mix. Over the last ten years, the capacity of DW installed in the U.S.

grew 10% on average annually (Sheridan et al., 2024). During that time, the U.S. Department of Energy (DOE) has made

continued investments in developing, certifying, and commercializing DW technologies—awarding 30 companies more than

$18.5 million to improve DW interoperability, cost-competitiveness, and design (Nrel, 2024). This has driven gradual

reductions in the levelized cost of energy (LCOE) for DW, with LCOE conservatively projected to drop by more than 40%

across technology sizes by the end of the decade (compared to 2022) (NREL). This decade has also witnessed significant

activity in the small wind market, with several international turbine manufacturers entering the U.S. market, as well as new

domestic start-ups working towards product commercialization (Sheridan et al., 2024). These advancements and

investments, alongside federal initiatives providing customer-facing financial support and opportunities, position DW for

more widespread adoption. For example, the IRA allocates grant funding to the U.S. Department of Agriculture (USDA) for

underutilized technologies like DW through the Rural and Agricultural Income & Savings from Renewable Energy (RAISE)

Initiative. Under this initiative, in collaboration with the DOE, USDA aims to assist 400 individual farmers in deploying

smaller-scale onsite wind projects (Hallett, 2024; Parker et al., 2024). The Federal Energy Regulatory Commission's 2020

order enabling distributed energy resources to participate in wholesale electricity markets further offers compelling revenue

streams for potential DW projects (Tapio and Preziuso, 2024). With substantial momentum for continued industry growth,

more focus should be placed on building a skilled workforce in the DW energy sector to prepare for future deployments

supported by these initiatives effectively.

DW workforce development has received fragmented attention to date, with periodic efforts led by installers to increase the

workforce in response to sector growth (Parker et al., 2024).  Even so, the number of installers and service providers in the

DW industry is still limited, which could potentially hinder the pace of market growth (Garbe et al., 2024). In addition,

economically favorable locations for DW projects, which can create jobs, have a strong correlation with disadvantaged

communities facing social, economic, or environmental barriers that hinder access to resources and opportunities (Mccabe et

al., 2022). These combined challenges point to an opportunity space: working with minority-serving institutions (MSIs) and

non-traditional academic providers that support underrepresented demographics, especially those located in wind-favorable

areas, to help build a diverse and equitable DW workforce.

This paper showcases the first phase of the Diverse and Equitable Workforce in Wind Energy (DEWWind) project, which

identifies potential industry and academic collaborators for DW workforce development. The collaborator identification

process utilizes a replicable rubric to ensure an equitable selection of collaborators and prioritize MSIs and underrepresented

demographics in wind workforce development. The remainder of the introduction will discuss gaps in the DW workforce

landscape and provide more background on the DEWWind project approach to addressing workforce capacity and diversity

needs. Section 2 outlines the methodology used to develop the rubric, including background on energy equity and its

application to this work and the implementation of the rubric in spatial analysis software to produce results (i.e., potential

candidates for workforce development). Section 3 reviews the results of rubric implementation, followed by section 4, which

provides discussion and reflections.





## 1.1 Gaps in Distributed Wind Workforce Development

Both small- and large-scale DW installers and manufacturers have reported difficulty hiring qualified candidates in recent
years (Figure 1 and Figure 2), reflecting a broader challenge in finding qualified candidates, as well as connecting qualified
candidates to jobs, across DW industry segments (Orrell et al., 2023; Stefek et al., 2022). Gaining wind energy-specific skills
and work experience, as well as identifying positions aligned with candidate skills, were noted as primary drivers for this
challenge. An additional challenge is the geographic disconnect between where wind industry jobs are located and where the
potential workforce is willing to live (Stefek et al., 2022). The findings highlight a missing link between wind industry
employers, the potential workforce, and educational institutions in building and connecting qualified and skilled career-
seekers to compatible wind jobs.

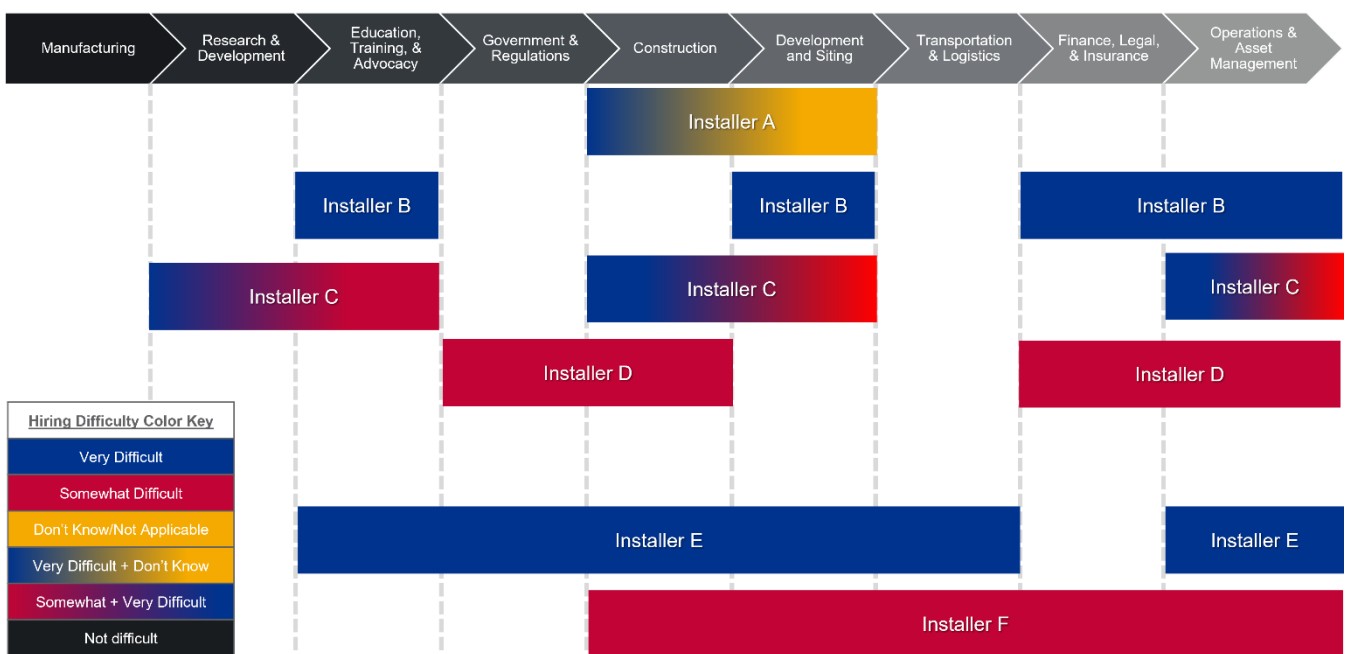

**Figure 1: Data gathered for the 2022 Distributed Wind Market Report revealed that most DW installers have difficulty hiring**
**across all industry segments. For example, Installer D worked across government and regulations, construction, finance, and**
**operations segments and reported finding hiring somewhat difficult.**





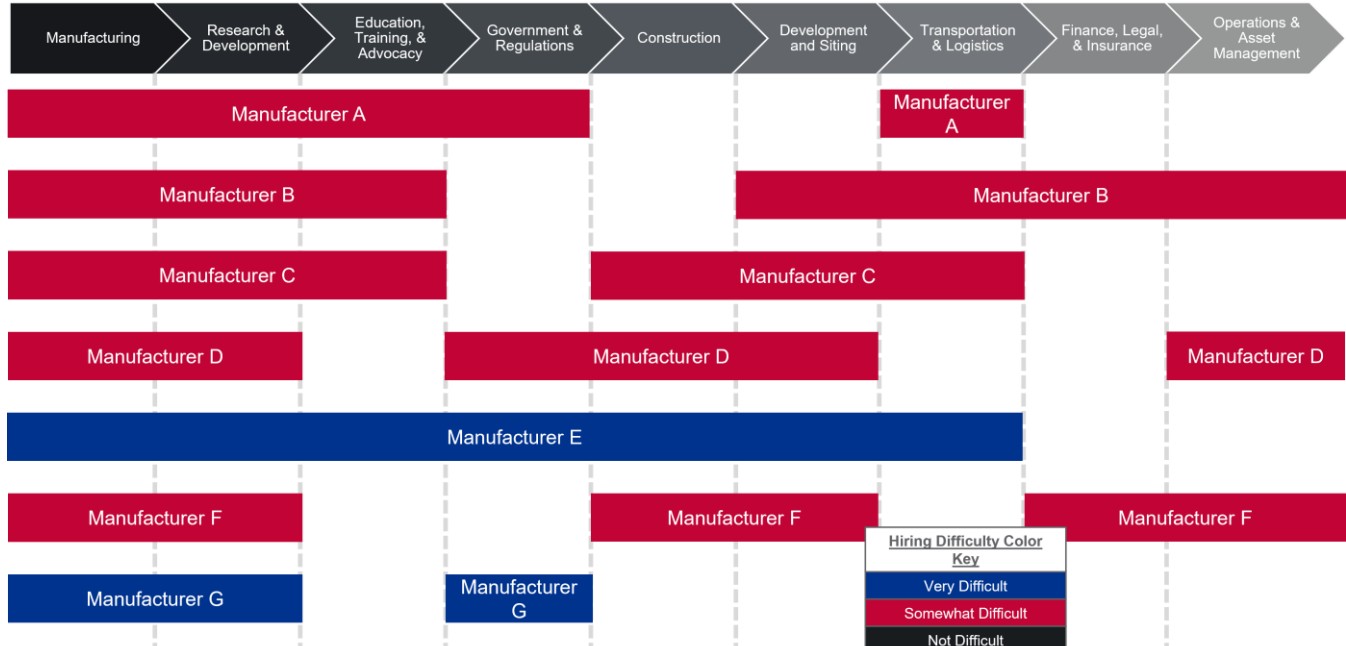

**Figure 2: Data gathered for the 2022 Distributed Wind Market Report revealed that most DW manufacturers had difficulty hiring across all industry segments, as evidenced by each company's blue or red shading.**

An additional gap in DW workforce development is the lack of overarching training and certification programs for the sector. Unlike land-based and offshore wind sectors, workforce development has been a relatively new objective for DW—not centralized or undertaken by an organizing state or federal agency. There are dedicated and specialized university programs, state-run training services, and accreditation boards for land-based and offshore wind, such as NYSERDA's offshore wind training institute and DOL-approved apprenticeship programs. However, DW efforts to date have been ad-hoc

and administered by key industry players in its limited network. For instance, the North American Board of Certified Energy Practitioners (NABCEP) brought together a group of small wind experts comprised of educators, installers, and other experienced wind energy leaders in 2010 to develop a Small Wind Associate Certification (NABCEP, 2010). By January 2012, nine candidates had received certification, but the program was indefinitely suspended as of September 2012, and no new applications were accepted (Oteri and Sinclair, 2012; NABCEP, 2018).

Overall, the wind energy sector has a below-average representation of marginalized groups—and the transition to a cleaner energy future is an opportunity to change that. To support a just and equitable clean energy transition, job creation and workforce development opportunities must be distributed fairly and encourage participation from communities currently under-represented in or under-served by the energy sector. The wind workforce is currently disproportionately ~70% male compared to the U.S. average of ~53%, and the representation of Black, African American, Asian, and individuals with

disabilities is lower than the national average (Mcdowell et al., 2024). This highlights an opportunity for DW workforce development to increase the number of qualified workers and align local job creation and economic benefits to underserved



and underrepresented communities. This opportunity to connect with communities is synergistic with the nature of DW being a site-specific, place-based energy generation resource that serves local customers and loads. This is a key opportunity area for DW as opposed to utility-scale wind that generates bulk power transferred over long distances, geographically

disconnecting the power from the communities in which it is produced.

A recent effort to develop a workforce roadmap for the DW sector defined two goals essential for initiating workforce development (Parker et al., 2024). Goal 1 is to increase interest, awareness, and visibility of the DW industry through new recruitment methods. Goal 2 is to meet the near-term need for multifunction workers while planning for long-term diversity of positions by identifying programs addressing distinct skill needs. With so much industry and workforce growth poised for

DW, there is an opportunity to align these roadmap solutions with DEWWind to support industry needs and more diverse and equitable workforce outcomes.

**1.2 DEWWind Approach**

DEWWind aims to strategize pathways for increased workforce diversity and support curricula-building for workforce development programs via industry and institutional collaboration. This includes working with MSIs, community colleges,

and non-traditional academic providers to reach students from underrepresented and disadvantaged backgrounds to help drive interest in DW careers and highlight visibility for various career opportunities. It also means collaborating with industry leaders to consider novel recruitment strategies and drive practical program-building responsive to the gaps the industry is currently seeing. Regional partnerships between academic institutions and industry leaders are a cornerstone of the DEWWind approach. The intention behind building regional partnerships aims to facilitate connections among

geographically proximate entities that can establish a positive feedback loop, ensuring a synergistic relationship between DW industry employers and educational programs critical to DW.

For sustainability, maximum sector impact, and advancement of diversity and equity objectives, the collaborator selection approach requires a robust and replicable methodology that concurrently centers the sector's needs and opportunities, as well as the project objectives. Relative to the wind industry as a whole, the DW sector is small (with roughly 1.1 GW of installed

capacity at the time of writing (Sheridan et al., 2024). This results in a limited sample size for industry partners; industry selection criteria can be defined by interest and availability. However, the potential DEWWind academic partners are in the thousands and thus require a more strategic selection method. Utilizing quantifiable selection criteria supports a more rigorous, fair, and effective partnership process. A quantifiable methodology minimizes bias, ensuring decisions are based on measurable data rather than subjective opinions. It allows for a standardized evaluation process, promotes transparency in

decision-making, makes it easier to justify selection decisions, and ensures alignment with the project objectives. Because of this, DEWWind utilized an equity-driven rubric that prioritizes academic organizations supporting underserved groups in rural, wind-rich communities to create equitable partnership opportunities in critical workforce development areas.

DEWWind has two direct value streams for potential partners: academic collaborators receive hands-on curriculum-building through program development informed by industry technical expertise, and industry collaborators benefit from accelerated



workforce development that plays into hiring needs across various industry segments. Direct collaboration, education, and
technical expertise are combined to address local and regional needs. Overall, the project develops a framework for outreach,
engagement, and program development that increases market readiness for accelerated DW deployment through equitable
workforce growth.

## 2 Materials and Methods

The DEWWind project seeks to bridge the workforce gap in the DW industry by fostering partnerships between academic
institutions and industry leaders to facilitate equitable outcomes in workforce development. This section outlines the
materials and methods for developing the DEWWind rubric, prioritizing education providers supporting underserved
communities in wind-rich areas. The methodology ensures a replicable and transparent selection process, centering on equity
to enhance workforce diversity in the DW sector. The following sub-sections detail the equity priorities, scoring criteria, and
spatial analysis techniques employed to accomplish the project's objectives.

### 2.1 Equity Priorities

As mentioned, the first phase of the DEWWind project was geared toward identifying and advancing new and equitable
partnership opportunities with education providers and industry leaders. There is tremendous potential to increase the
number of wind energy workers and, more importantly, the diversity of the DW energy workforce by engaging MSIs and
technical and trade schools, especially those located in areas favorable for DW deployment. This can also support local
economic development since high wind resource quality areas can often be in remote, economically distressed communities.
Disadvantaged communities represent 47% of all parcels where behind-the-meter DW applications can be sited and 43% of
all parcels where front-of-the-meter DW applications can be sited within the contiguous United States (Mccabe et al., 2022).
Further, the Midwest, Heartland, Northeast, and portions of the Mountain West regions where DW's economic potential is
high intersects with swaths of rural America (Mccabe et al., 2022). Identifying partnership opportunities capitalizes on these
correlations through specific equity priorities that ensure collaboration with academic organizations supporting underserved
groups.
Four equity priorities were defined to prioritize collaboration with academic organizations supporting underserved groups in
wind-rich communities. These priorities aim to enhance collaboration with academic organizations that support underserved
groups, thereby addressing systemic barriers and fostering diversity within the workforce. Below are the specific priorities:

1.   **DEWWind prioritizes currently underserved or underrepresented groups in the DW industry.** As defined in
Executive Orders 13985, 14020, and 14091, the term "underserved communities" refers to those populations as well
as geographic communities that have been systematically denied the opportunity to participate fully in aspects of
economic, social, and civil life and may include Black, Latino, Indigenous and Native American, Asian American,
Native Hawaiian, and Pacific Islander persons and other persons of color; members of religious minorities; women



and girls; LGBTQI+ persons; persons with disabilities; persons who live in rural areas; persons who live in United States Territories; persons otherwise adversely affected by persistent poverty or inequality; and individuals who belong to multiple such communities. Underserved communities also include individuals with limited proficiency in English, whether they use spoken language, sign language, or other communication methods per Executive Order 14094. The energy sector has a below-average representation of Hispanic or Latinx workers and Black or African American workers and a below-average proportion of women ( Research Partnership, 2021).

2. **DEWWind prioritizes MSIs, community colleges, and technical and trade programs.** MSIs align with equity priority 1, while community colleges and technical and trade programs are often dedicated to skilled job training, such as those required for the DW workforce.

3. **DEWWind prioritizes rural areas due to high DW deployment potential and unique energy equity considerations for rural loads.** Rural areas represented a significant percentage of newly installed U.S. DW projects deployed in 2022 (Orrell et al., 2023). Consumers with rural energy loads are more likely to have a higher energy burden, experience more significant grid reliability challenges, and be exposed to more aging and inefficient grid infrastructure than their metropolitan counterparts (Parker et al., 2023).

4. **DEWWind prioritizes institutions within 100 miles of active installers. "**Active" installers are defined as having at least three or more projects in the last five years (Orrell et al., 2023). DEWWind focuses on installers rather than manufacturers because installers represent part of the project cycle segment that needs expansion to meet increased demand for DW in the future. Being place-based by nature, DW needs a local workforce connected to installers to service projects. The proximity radius is applied to ease travel needs and collaboration once partnerships are established while also addressing the challenge of the geographic disconnect between the locations of wind industry jobs and the areas where the potential workforce is willing to reside.

## 2.2 Rubric Development

Rubric development builds on the equity priorities by incorporating weighted locational, institutional, and socioeconomic criteria that align with the project's equity objectives. Weighting is not meant to assign a rank to potential collaborators nor act as a precise measure for determining suitability; instead, it illuminates academic organizations with favorable characteristics for DW workforce development aligned with the project's objectives. Each rubric criterion is framed through the lenses of procedural and recognition justice. Procedural justice looks at the fairness of decision-making processes, ensuring participants can define, drive, and hold accountable program decisions and outcomes. Recognition justice emphasizes the need to understand different vulnerability types and specific needs among social groups, especially marginalized communities. Both justice aspects apply transparency, accountability, and due process principles. Transparency brings about accountability by empowering people with information to hold institutions accountable and shed light on decision-making processes (Tarekegne et al., 2021; Lanckton and Devar, 2021).



Thus, in an effort to further these principles per procedural and recognition justice, we utilize this rubric as a measurable evaluation criterion (i.e., metrics) to make it easier to hold the project accountable in participant selection. The rubric

criterion is a combination of two types of equity metrics: target metrics and tracking metrics (Tarekegne et al., 2021). Target metrics capture descriptive analytics on populations and are demographic-specific measurements. They speak to recognition justice and will contribute to diverse workforce representation. Tracking metrics reflect progress measurement (i.e., program sustainability, self-ownership, longevity, etc.) and can evaluate how well an effort has helped a target community. They speak to procedural justice and how well workforce development programs address local perspectives. Tracking metrics will

further set appropriate, achievable equity-related goals to undo past disparities. Both types of metrics will inform collaborators of ways to increase program efficacy and reach.

Further, the rubric streamlines the evaluation of potential academic collaborators for DEWWind. Rubric criterion abstract procedural and recognition justice principles from their energy justice roots and re-align them towards diversity, equity, inclusion, and accessibility DEIA in order to achieve the equity priorities. The criteria categories align with recognition

justice. The methodology for collaborator evaluation aligns with procedural justice. Examples of other energy equity and justice work that uses a rubric for evaluation include the American Council for an Energy-Efficient Economy (ACEEE) scorecards (American Council for an Energy-Efficient Economy, 2024) and the "Justice in 100" scorecard from the Initiative for Energy Justice (Lanckton and Devar, 2021).

There are three criterion categories with a subset of metrics aligned with the equity priorities, as shown in Table 1.

- **Institution Type** considers what kind of academic institution the potential academic partner is. Because of Equity Priority 2, MSIs, community colleges and technical and trade programs, and women's colleges receive 5, 3, and 2 points, respectively. Institutions not classified in these three designations get 1 point.

- **Location** considers where the institution is located. Because of Equity Priority 4, institutions within 100 miles of the installer get 3 points. Because of Equity Priority 3, areas classified as rural per the US Department of

Agriculture's Rural Energy for America Program (REAP)[1] eligibility get 2 points. Institutions in or near wind-rich areas get 1 point. Wind richness is defined per the Distributed Wind Energy Futures Study (Mccabe et al., 2022) through capital expenditure thresholds.[2] Though breakeven costs do not necessarily capture all "wind-rich" locations, areas above the 80th percentile, along with a combination of other factors, are considered economically favorable for DW deployments (see Table 3).

- **Demographic and Socioeconomic Indicators** characterize the disadvantages in the census tract where the institution is located. Because of Equity Priority 1, institution census tracts with the aggregate minority population

---

[1] All locations not in "ineligible areas" meet USDA's definition of rural for REAP applications, which is a target funding source for the RAISE initiative.
https://eligibility.sc.egov.usda.gov/eligibility/welcomeAction.do;jsessionid=sbaz4pqebSEqobTswlZfSdIM
[2] Threshold CapEx is an indicator of the amount of capital that could be invested for a system at a specific site while still maintaining profitability; higher threshold CapEx values mean higher favorability for DW energy.





at or above the 75th percentile get 3 points. Minority status is determined by the Environmental Protection Agency's EJSCREEN tool, which helps identify areas with environmental burdens and vulnerable populations (United States Environmental Protection Agency, 2024a). Because of the overall goals of this work, workforce development

disadvantage indicators are worth 2 points. This includes linguistic isolation, low median income, poverty level, an unemployment rate at or above the 90th percentile, and high-school education above 10%. These indicators are explained in further detail in the overview of socioeconomic indicators for EJSCREEN (United States Environmental Protection Agency, 2024b).

**Table 1: DEWWind's collaborator selection criterion is utilized in a weighted rubric aligned with equity priorities in Sec. 2.**

| Category | Criteria | Points Awarded | Equity Priority |
|---|---|---|---|
| **Institution Type** | Minority-Serving Institutions (MSIs), e.g., historically black colleges and universities, tribal colleges, etc. | 5 | 1, 2 |
| | Community colleges and technical and trade Institutions, i.e., technical colleges, trade schools | 3 | 2 |
| | Women's colleges and universities | 2 | 1 |
| | All other colleges or universities not classified by the above designations | 1 | N/A |
| **Location** | Institution within 100 mi of DW installer | 3 | 4 |
| | Institution in rural areas per USDA REAP eligibility | 2 | 3 |
| | Institution in wind-rich areas with behind-the-meter (BTM)/front-of-the-meter (FTM) DW capital expenditure at or above the 80th national percentile[1] | 1 | N/A |
| **Demographic and Socioeconomic Indicators** | Institution census tract with aggregate minority population at or above the 75th national percentile | 3 | 1 |
| | Institution census tract with "less than high school education" population at or above 10% | 2 | 1 |
| | Institution census tract with low-income population at or above the 90th national percentile | 2 | 1 |
| | Institution census tract with "limited English speaking" populations (linguistic isolation) at or above the 90th national percentile | 2 | 1 |
| | Institution census tract with unemployment at or above the 90th national percentile | 2 | 1 |






The scoring formulas were applied to every academic institution and technical and trade school in the United States. The highest theoretical score possible is 25, a case in which an academic institution would be awarded 8 points for qualifying as a minority-serving community college (5 points for MSI type; 3 points for community college institution type), 3 points for being within 100 miles of a DW installer, 2 points for being located in a rural area, 1 point for being located in a wind-rich

area, and 11 points for meeting all socioeconomic criteria thresholds. The higher the score, the more likely the institution satisfies the project objectives and equity priorities.

**2.3 Spatial and Mapping Implementation**

A Geographic Information System (GIS) combined with RStudio was used to score all post-secondary education institutions, with the list of colleges and universities (C&U) supplied by HIFLD (HIFLD, 2020) and based on the scoring rubric outlined

in Section 2.2.  Institution types (e.g., MSI, community college) were pre-labeled within this data layer. First, the MSI institutions were read in (NASA, 2024). The *left_join()* function from *dplyr* packages combined both datasets based on address fields. The updated C&U data was then read into Arc GIS Pro as *xy* data. The C&U layer was then spatially joined[3] with demographic and socioeconomic indicators from EJSCREEN (Table 2).

**Table 2: Demographic value thresholds derived from R extraction of all US Census Tracts.**

| Demographic value | Threshold | Value |
|---|---|---|
| Minority | 75th percentile | 0.633 |
| Low income | 90th percentile | 0.572 |
| Unemployment | 90th percentile | 0.116 |
| Linguistically isolated | 90th percentile | 0.134 |
| Less than high school education | 10% | 0.100 |


For the location criterion, rural status, defined by USDA REAP eligibility, was spatially joined to the C&U layer as target features with *intersect* match option.  To assess proximity to DW installers, point locations of institutions and addresses of installer headquarters were geo-located. With C&U as input features, we select by location with "Relationship" as "Within a distance," "Selecting Features" as the installer point locations, and "Search Distance" as 100 "US Survey Miles." We then

added a new field to the C&U layer as a yes/no to installer proximity. Next, the wind-richness data was added to the C&U layer by spatially joining *dWind* data, which considers the front-of-the-meter and behind-the-meter CapEx thresholds. To get the respective thresholds for these attributes according to the scoring rubric, the Python Pandas library was used to extract those values from the entirety of the Distributed Wind Energy Futures Study (Table 3).

---

[3] Spatially joined refers to the process of combining two datasets based on their geographic relationship or spatial proximity, rather than their attributes alone. This means that features from one dataset are linked to features in another dataset based on their locations (e.g., points, lines, or polygons) within a defined spatial area.



**Table 3: CapEx Thresholds derived from GIS Outputs based on Distributed Wind Energy Futures Study (Mccabe et al., 2022)**

| CapEx Criteria | Threshold | Value ($/kW) |
|---|---|---|
| Front-of-the-meter | 80th | 1180 |
| Behind-the-meter | 80th | 5881 |

This GIS analysis resulted in a single CSV file containing institutional, locational, and socioeconomic scores for all C&U. These CSV files were converted to Excel spreadsheets and combined for post-processing, which included manually scoring the 25 women's colleges, removing "specialized" educational institutions, such as performing arts schools, cosmetology schools, and seminaries, and filtering out academic institutions located outside the contiguous U.S., for which there are multiple data gaps (e.g., no CapEx data is available for AK, HI, and U.S. territories). The final step in post-processing was

validating the GIS results by embedding formulas in the spreadsheet to verify the final scores.

## 3 Results

The preliminary pre-processed dataset contained 6,839 institutions. After pre-processing to remove flight training, cosmetology and barber, fine arts, and educational support programs and manually adding points for women's colleges, the final dataset contained 5,106 post-secondary institutions with scores ranging from 1 to 23 with a mean of 7.5 and median of

7. Because of the active installer criteria, i.e., at least three or more projects in the last five years, fewer than 20 installers are included in the results. They also reflect the highly responsive DW industry partners known to be interested in supporting the DEWWind project. There were 25 women's colleges, 1,538 junior and community colleges, 1,034 technical and trade schools, 794 MSIs, and 2,102 other institutions. Figure 3 shows the resulting scores for the schools compared with installer locations. Alaska and Hawaii are not included in the results due to unreliable data on wind-richness.

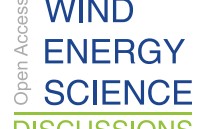

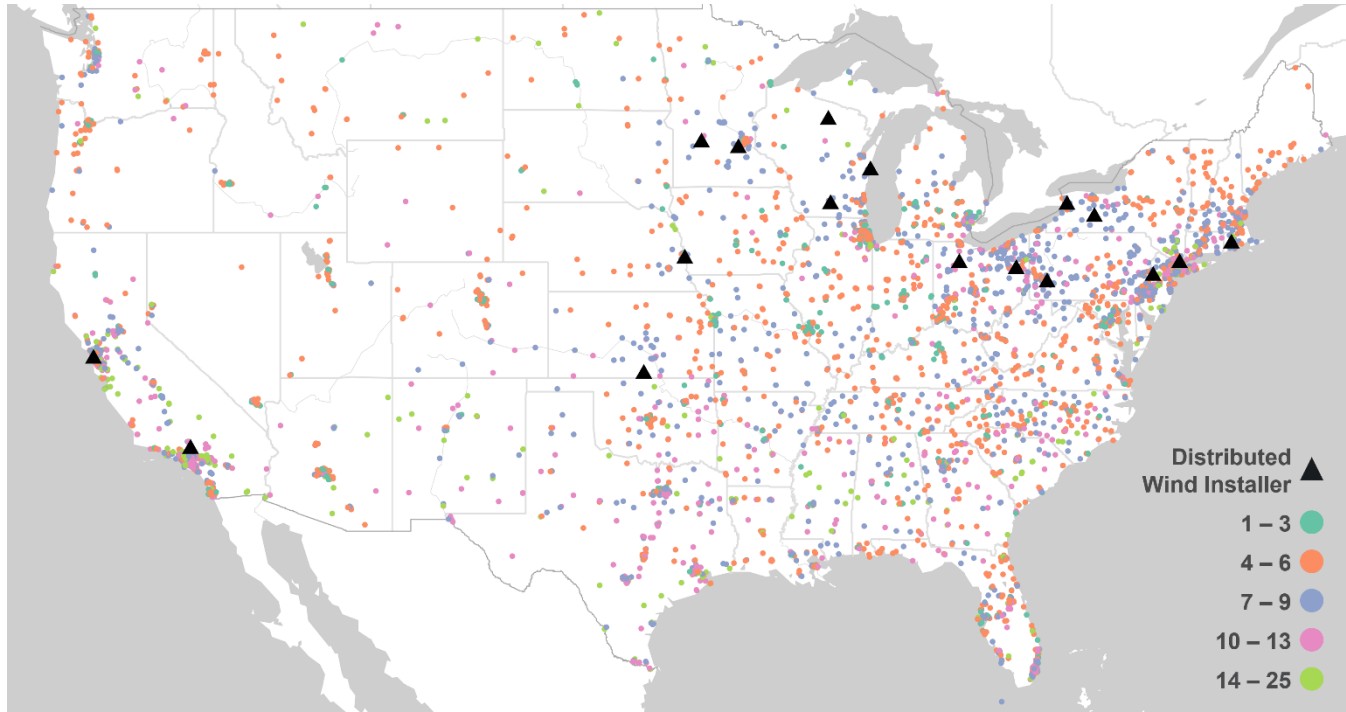

**Figure 3: Final score map for all institutions in the contiguous US with installers included.**

## 4 Discussion and Reflections

A regional and institutional assessment gives insight into the distribution of scores. Figure 4 shows a breakdown of scores by institution type. In regions such as the Southwest (SW) and Southeast (SE), there are many high-scoring schools that are MSIs (Figure 4a) with scores of 10 or above but no nearby installers. This discrepancy indicates a potential challenge in aligning high-scoring academic institutions with local industry needs. Institutions located in regions characterized as rural by USDA REAP criteria scored higher due to their alignment with equity priorities. Additionally, areas that are wind-rich gave a small geographic advantage to institutions in these regions. Institutions within 100 miles of an installer (Figure 4b) primarily scored in the 4 to 9 range, indicating the small impact of the proximity criteria on final scores. Institutions with scores 20 or above achieved those scores by fulfilling all socioeconomic and demographic criteria in addition to the maximum institution points (Figure 4d).



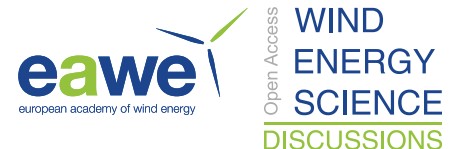

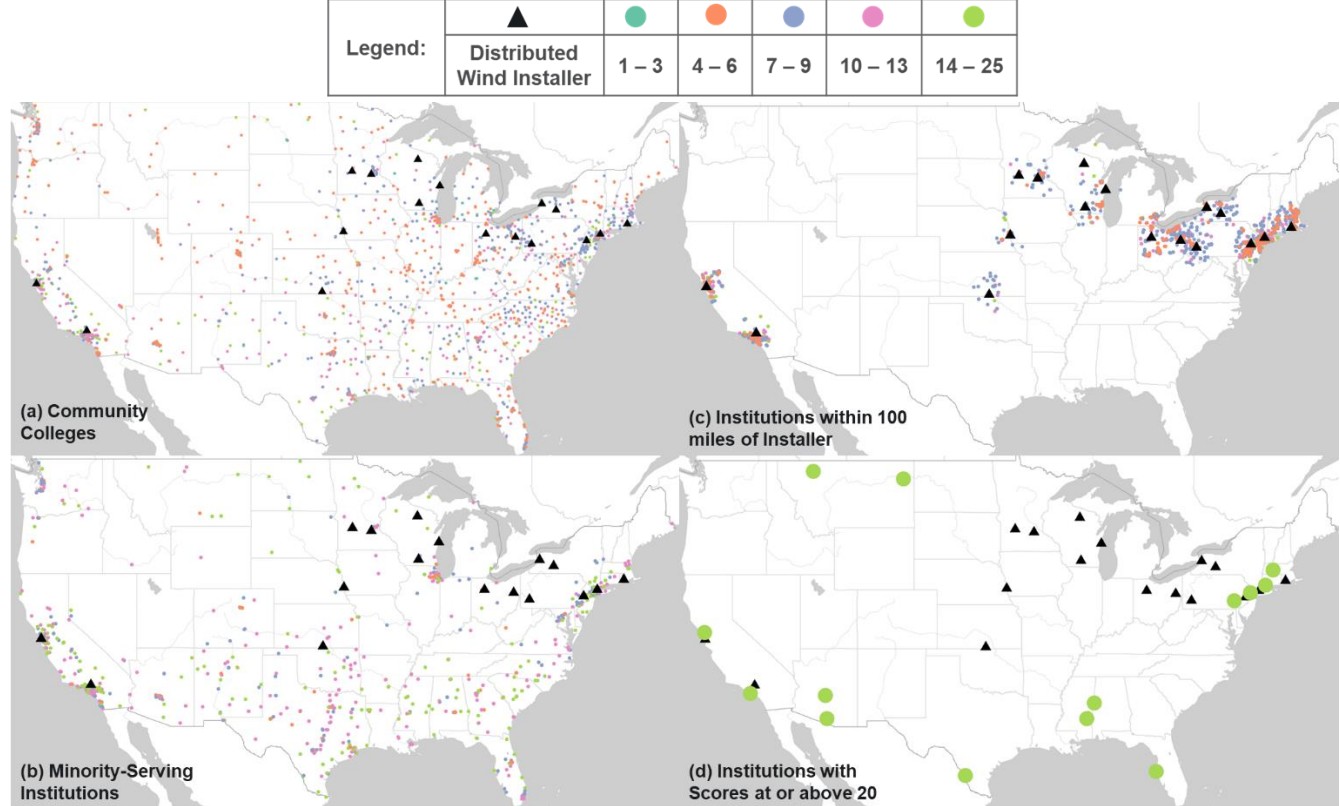

**Figure 4: Scores for (a) junior and community colleges, (b) MSIs, (c) institutions that meet the installer proximity criteria, and (d) institutions with scores above 20.**

A further assessment of the score frequencies in Figure 5a shows a skewed right tail distribution indicating the majority of institutions scored below 9 with outliers above 15 and below 2. The right-skew suggests that while the majority of institutions have limited alignment with equity priorities, there are outliers where institutions score much higher. The 3 to 9 score range had the highest frequency with about 65% of institutions falling in this interval reflecting the institutions with low socioeconomic and demographic scores and those with minimal points in the institution category. Scores above 15 are mostly MSIs and junior colleges with high scores in the location and socioeconomic categories.

The violin plots (Figure 5b) are variations of the traditional box-and-whisker plots that provide insights into the variability of scores within each institutional category. The median scores for MSIs are notably the highest, likely reflecting these institutions' long-standing commitments to supporting underrepresented communities. The rubric weights capture this characteristic by giving 5 points to MSIs. In contrast, junior and community colleges exhibit the widest range and greatest variability in scores, which may reflect the diversity of student populations and resources available across different colleges. Institutions that do not fit into the MSI, junior college, women's college, or trade/technical school categories had the most outliers, with some institutions showing exceptional performance against the rubric, while others did not.



The multi-modal nature of the distributions across all institutional categories suggests that each group has varied characteristics that would make this suitable for the DEWWind project. Some institutions may be well suited locationally but because of their student population they lose out on institution points. Conversely, some institutions dominated the institution criterion but may not be located in census tracts that fit the rubric's socioeconomic and demographic requirements. These visualizations help illustrate the rubric's performance disparities and aid future refinement of the collaborator selection approach.







**Figure 5: The distribution of scores illustrated by (a) a histogram and (b) violin plots showcasing the interquartile ranges and distribution shapes by institution type. Note that total counts exceed 5,106 due to institutions falling in multiple categories.**





In evaluating the rubric's success in meeting project objectives, we can consider if the rubric effectively prioritized institutions that support underserved and underrepresented communities, particularly in rural, wind-rich areas. The highest-scoring institutions do reflect these equity priorities. And, after performing outreach to these top-scoring institutions, those selected for final partnerships reflect a mix of institution types with varying geographic, institutional, and socioeconomic profiles.

However, there is inherent tension and trade-offs in optimizing objectives. While the project aims to prioritize wind-rich areas, underrepresented institutions, and proximity to DW installers, achieving balance remains challenging. For example, some high-scoring institutions might not be located near DW installers. Rural areas and the Midwest and Northeast regions had institutions with scores above 20 that best balanced DEWWind's objectives. The small number of active installers (i.e., underdeveloped market) relative to institutions that meet some of the equity priorities influences this tension and reflects a

challenge given the state of the industry. However, it also points to a gap and future research area that can refine the rubric's criteria.

Given that the DW industry network is relatively small—with many key stakeholders already over-taxed through involvement in other DOE-based R&D efforts—the results point to new connections and partnership opportunities that can broaden DOE's overall network. Leveraging workforce efforts for utility-scale or offshore wind is an option to expand

partnerships but it demands considerable financial resources, staff time, and infrastructure, which DW companies might find challenging to secure. In addition, the DW sector boasts a multifunctional worker model requiring employees with broad abilities that are difficult to translate to the wind industry at large (Parker et al., 2024). Application of DEWWind's rubric leverages the relatively small and overstretched DW industry network to locate new stakeholders that align with the project's equity and workforce development objectives.

The application of the equity-driven rubric can serve as a strategic tool to identify and engage academic institutions and vocational programs in wind-rich, underserved areas that currently lack nearby DW installers. By prioritizing MSIs and community colleges, particularly those scoring high on the rubric but lacking nearby installers, the DEWWind project can foster local workforce development, tailored curriculum-building, and strategic partnerships. This can potentially attract new installers to these regions by highlighting untapped market opportunities and demonstrating a ready and diverse workforce.

These efforts, in turn, could motivate DW companies to expand their operations into these high-scoring areas, ultimately increasing the number of installers and developers in regions currently underserved by the industry.

With the collaborator analysis complete, the next steps for the DEWWind project are to initiate outreach with the highest-scoring institutions and the installer in the closest proximity. After initial outreach—and once industry and academic collaborators have confirmed their interest in participation—PNNL will work with collaborators to address workforce

development needs through workshopping events that will ultimately inform workforce program development. Although small in scale, these collaborative opportunities will hopefully highlight a way to scale up equitable partnerships to address DW workforce needs more comprehensively.



**Code and Data Availability**

The data used is publicly available and can be accessed through the corresponding citations. The code is deposited in Wind

Energy Science Journal's FAIR-aligned data repository.

**Author Contribution**

Dr. Kendall Parker designed rubric and prepared the manuscript with contributions from all co-authors. Danielle Preziuso and Micah Taylor developed the code and performed the spatial and mapping implementation.

**Competing Interests**

The authors declare that they have no conflict of interest.

**Acknowledgements**

The authors would like to thank the Department of Energy Wind Energy Technologies Office for their financial support of this work.

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
