# Peer review of "Building a Diverse and Equitable Distributed Wind Workforce: A Strategic Approach to Collaborator Selection"

_Wind Energy Science, 2024_

## Referee Comment (RC2)

**Building a Diverse and Equitable Distributed Wind Workforce: A Strategic Approach to Collaborator Selection**

Supplement - Feedback

Line 10 – Suggest looking to the U.S. Energy and Employment Report for wind energy demographic data. Discuss what improvements distributed wind could provide.

Line 55 – I would like to see a discussion if distributed wind has specialized skills (or different from utility scale LBW) that may impact industry and workforce development for these programs.

Line 85 – I'm confused why Development and Siting is after listed after Construction in Figures. Consider putting more professional type phases next to each other?

Figure 2 – How was the question asked to manufacturers different than installers? How much insight would a manufacturer have into construction hiring or development?

Line 115 – Adding a discussion about distributed wind energy demand (current and future employment) may strengthen this discussion. Even empirical information on how many folks are in an installation crew or how many folks are needed for different deployment phases would help add context.

Line 109 – Do you have any information on how different underserved and underrepresented communities could be from those communities where utility-scale LBW is constructed? Present different economic or demographic data. While the power is transferred somewhere else, is it generated at similar locations. How demographically diverse are these communities? Rural populations are underserved and underrepresented communities, but their workforce development may continue to have a smaller impact on the worker demographics. Which parts of increasing diversity and equity would this project has the biggest influence?

Line 110 – what is the definition of "multifunction workers" – workers trained across job or sectors with overlapping skillsets?

Line 125 – is the project objectives DEWWind – maybe add that explicitly?

Line 165 – list dates or administration of EO?

Line 175 – How are registered apprenticeship programs considered? Do you prioritize RAPs to comply with IRA PWA (since it was mentioned as important in the intro?)

Line 240 – Were institutions filtered to include those programs that had a program type that had relevant skillsets for distributed wind energy? If a program has an existing wind programs, how does that apply? Meaning, if there was an existing wind training program within a particular area, should it get plus points?

Line 295 – indication that training development needs to be around installers; but could you also run a specific analysis to understand training program expansion or development for new market areas?

---

## Author Comment (AC2)

**Building a Diverse and Equitable Distributed Wind Workforce: A Strategic Approach to Collaborator Selection**

Supplement - Feedback RC 2

Overall - The Building a Diverse and Equitable Distributed Wind Workforce: A Strategic Approach to Collaborator Selection provided a novel and reasonable approach to identifying training programs that can increase diversity and equity in wind energy workforce through distributed wind technology. As an experienced wind workforce analyst, the methods are sound and provide valuable approach for making data driven and informed workforce development decisions. **To strengthen the article, the authors may consider a more detailed discussion on the types of underserved and underrepresented communities. For example, like utility-scale deployment, distributed wind is likely is occurring in more rural areas. What are the unique economic and demographic implications of these types of communities on increasing diversity and equity in wind energy.** Several minor comments for authors consideration are also attached as a supplement.

Thank you so much for your thorough and thoughtful feedback. We've made edits to the manuscript to address your supplemented comments and added discussion to section 1.1 on the economic and demographic implications of rural, underserved, and underrepresented communities on wind energy.

In addition, per guidance from our sponsor the project name has been changed from Diverse and Equitable Workforce for Wind (DEWWind) to **Workforce Innovation for Distributed Wind Advancement, Recruitment, and Development (WINDWARD).** The comment responses below and the updated manuscript will reflect this name change.

Line 10 – Suggest looking to the U.S. Energy and Employment Report for wind energy demographic data. Discuss what improvements distributed wind could provide.

Thank you for suggesting this resource. We've added some insights that reflect the findings in the USEER.

Line 55 – I would like to see a discussion if distributed wind has specialized skills (or different from utility scale LBW) that may impact industry and workforce development for these programs.

We've added a few sentences of discussion on the unique skills distributed wind requires compared to LBW. A key piece of future work for the WINDWARD project is to further map and define these skills. So far, previous work has identified that DW needs multifunction workers - employees with broad abilities that can perform various tasks, from installation and maintenance to troubleshooting and customer service.[1]

Line 85 – I'm confused why Development and Siting is after listed after Construction in Figures. Consider putting more professional type phases next to each other?

The top heading bars incorrectly illustrate the industry segments as if they are sequential. We've updated the graphic to make the industry segments more individualized instead of them seemingly depicting a process. The order of the segments reflects the selection order in the survey question.
* * *
[1] Parker, Kendall M., Esaki-Kua, Lauren A., & Preziuso, Danielle C. (2024). Towards a Workforce Roadmap for Distributed Wind: Phase 1 - Identifying Needs and Barriers. https://doi.org/10.2172/2440158

Figure 2 – How was the question asked to manufacturers different than installers? How much insight would a manufacturer have into construction hiring or development?

The question was identical for both. There are manufacturers that identified themselves as also conducting turbine construction.

Line 115 – Adding a discussion about distributed wind energy demand (current and future employment) may strengthen this discussion. Even empirical information on how many folks are in an installation crew or how many folks are needed for different deployment phases would help add context.

Thank you for this suggestion. We've added additional context.

Line 109 – Do you have any information on how different underserved and underrepresented communities could be from those communities where utility-scale LBW is constructed? Present different economic or demographic data. While the power is transferred somewhere else, is it generated at similar locations. How demographically diverse are these communities? Rural populations are underserved and underrepresented communities, but their workforce development may continue to have a smaller impact on the worker demographics. Which parts of increasing diversity and equity would this project has the biggest influence?

Thank you for this thoughtful comment. We've made a few updates in this section to incorporate your suggestions.

- Added examples of LBW and DW turbine to corroborate the locality claims
- Started a new paragraph to highlight that the "local loads and local workers" contribution is advances equity
- Clarified the previous paragraph to explain that WINDWARD major contribution is to increasing diversity

Line 110 – what is the definition of "multifunction workers" – workers trained across job or sectors with overlapping skillsets?

This definition was added to address your comment at line 55 and reiterated in the manuscript at line 110 for clarity.

Line 125 – is the project objectives DEWWind – maybe add that explicitly?

We're clarified the connection between the WINDWARD project objectives and approach showcased in this work.

Line 165 – list dates or administration of EO?

We've removed mention of these Executive Orders per our sponsor's guidelines.

Line 175 – How are registered apprenticeship programs considered? Do you prioritize RAPs to comply with IRA PWA (since it was mentioned as important in the intro?)

RAPs are not given extra weight beyond what is already specified for technical and trade programs. We have added additional context to clarify this.

Line 240 – Were institutions filtered to include those programs that had a program type that had relevant skillsets for distributed wind energy? If a program has an existing wind programs, how does that apply? Meaning, if there was an existing wind training program within a particular area, should it get plus points?

No, institutions were not pre-filtered based on overlap with distributed wind skillsets because no current resource exists that identifies the necessary skills for a DW worker. In addition, existing wind programs did not get additional points because of WINDWARD's objective to build up new programs and curriculum (1) through partnerships with new institutions and (2) in areas with predominantly underserved and underrepresented groups.

Line 295 – indication that training development needs to be around installers; but could you also run a specific analysis to understand training program expansion or development for new market areas?

This is a great suggestion and aligns well with the next steps of WINDWARD to use the place-based characteristics near academic-industry partnerships to assess potential workforce program development. We've added more detail on these next steps.

---

## Author Response (AR1)

**Building a Diverse and Equitable Distributed Wind Workforce: A Strategic Approach to Collaborator Selection**

Author's Response

Only reviewer #2 provided written comments requiring edits to the manuscript. Those comments are addressed in this document. Responses to reviewer comments (RC) are in *blue italics*.

**Feedback RC 2**

The Building a Diverse and Equitable Distributed Wind Workforce: A Strategic Approach to Collaborator Selection provided a novel and reasonable approach to identifying training programs that can increase diversity and equity in wind energy workforce through distributed wind technology. As an experienced wind workforce analyst, the methods are sound and provide valuable approach for making data driven and informed workforce development decisions. To strengthen the article, the authors may consider a more detailed discussion on the types of underserved and underrepresented communities. For example, like utility-scale deployment, distributed wind is likely is occurring in more rural areas. What are the unique economic and demographic implications of these types of communities on increasing diversity and equity in wind energy. Several minor comments for authors consideration are also attached as a supplement.

**Response**

Thank you so much for your thorough and thoughtful feedback. We've made edits to the manuscript to address your supplemented comments and added discussion to section 1.1 on the economic and demographic implications of rural, underserved, and underrepresented communities on wind energy.

In addition, we have removed the project name from the manuscript and refer to the approaches in this work more generically per internal guidance. The comment responses below and the updated manuscript will reflect this name change.

**Supplement - Feedback RC 2**

Line 10 – Suggest looking to the U.S. Energy and Employment Report for wind energy demographic data. Discuss what improvements distributed wind could provide.

Thank you for suggesting this resource. We have added some insights that reflect the findings in the USEER but do not reference them formally because this is the abstract. There are additional references to this report in the introduction.

Line 55 – I would like to see a discussion if distributed wind has specialized skills (or different from utility scale LBW) that may impact industry and workforce development for these programs.

We have added a few sentences of discussion on the unique skills distributed wind requires compared to LBW. So far, previous work has identified that DW needs multifunction workers - employees with broad abilities that can perform various tasks, from installation and maintenance to troubleshooting and customer service. A key piece of future work for the workforce effort is to further map and define the skills needed for a DW workforce, and additional context has been added in the following paragraph to give background on the project's full scope.

Line 85 – I'm confused why Development and Siting is after listed after Construction in Figures. Consider putting more professional type phases next to each other?

The order of the segments reflects the selection order in the original survey questions. However, the top heading bars incorrectly illustrate the industry segments as if they are sequential. Both Figure 1 and Figure 2 have been updated to make the industry segments more individualized instead of them seemingly depicting a process.

Figure 2 – How was the question asked to manufacturers different than installers? How much insight would a manufacturer have into construction hiring or development?

Manufacturers and installers were provided identical survey questions. There are manufacturers that identified themselves as also conducting turbine construction and thus provided responses for the construction and development segments. I have added a footnote with this detail to the figure references in-line.

Line 115 – Adding a discussion about distributed wind energy demand (current and future employment) may strengthen this discussion. Even empirical information on how many folks are in an installation crew or how many folks are needed for different deployment phases would help add context.

Thank you for this suggestion. These numbers are challenging to gather because of the limited public information about the distributed wind workforce beyond the Distributed Wind Market Report.2 The report analyzes the U.S. distributed wind energy sector, and highlights key trends, market dynamics, and technological advancements. However, we have added additional context to support the claim of DW growth needing a growing workforce. In addition, the beginning of the introduction primes this discussion with policy motivation.

Line 109 – Do you have any information on how different underserved and underrepresented communities could be from those communities where utility-scale LBW is constructed? Present different economic or demographic data. While the power is transferred somewhere else, is it generated at similar locations. How demographically

<sup>1 Parker, Kendall M., Esaki-Kua, Lauren A., & Preziuso, Danielle C. (2024). Towards a Workforce Roadmap for Distributed Wind: Phase 1 - Identifying Needs and Barriers. https://doi.org/10.2172/2440158

<sup>2 Sheridan, L. M., Kazimierczuk, K., Garbe, J. T., & Preziuso, D. C. (2024). Distributed Wind Market Report: 2024 Edition. Pacific Northwest National Laboratory.

https://www.pnnl.gov/main/publications/external/technical\_reports/PNNL-36057.pdf

diverse are these communities? Rural populations are underserved and underrepresented communities, but their workforce development may continue to have a smaller impact on the worker demographics. Which parts of increasing diversity and equity would this project has the biggest influence?

Thank you for this thoughtful comment. We've made a few updates in this section to incorporate your suggestions. We started a new paragraph to highlight that the "local loads and local workers" contribution is advances equity, and workforce effort's contribution to this is in the process the project undertakes to contribute to more equitable futures. In addition, there is added clarity on contributions to diversity and their relevance to the wind industry.

Line 110 – what is the definition of "multifunction workers" – workers trained across job or sectors with overlapping skillsets?

This definition was added to address your comment at line 55 and reiterated in the manuscript at line 110 for clarity.

Line 125 – is the project objectives DEWWind – maybe add that explicitly?

Thanks for this suggestion. We added detail on the objectives of the workforce effort and clarified the connection between the objectives and approach showcased in this work.

Line 165 – list dates or administration of EO?

We have added publishing years to each of the Executive Orders.

Line 175 – How are registered apprenticeship programs considered? Do you prioritize RAPs to comply with IRA PWA (since it was mentioned as important in the intro?)

RAPs are not given extra weight beyond what is already specified for technical and trade programs. We have added additional context in this section and in the Rubric Development section to clarify this.

Line 240 – Were institutions filtered to include those programs that had a program type that had relevant skillsets for distributed wind energy? If a program has an existing wind programs, how does that apply? Meaning, if there was an existing wind training program within a particular area, should it get plus points?

No, institutions were not pre-filtered based on overlap with distributed wind skillsets because no current resource exists that identifies the necessary skills for a DW worker. In addition, existing wind programs did not get additional points because of the objective to build up new programs and curriculum (1) through partnerships with new institutions and (2) in areas with predominantly underserved and underrepresented groups. We have added details to this point under the Institution Type description.

Line 295 – indication that training development needs to be around installers; but could you also run a specific analysis to understand training program expansion or development for new market areas?

This is a great suggestion and aligns well with the next steps of the workforce effort to use the place-based characteristics near academic-industry partnerships to assess potential workforce program development. We have added more detail on these next steps.

---

## Referee Report (RR1)

This paper contributes to knowledge on how US workforce development efforts in distributed wind (DW) can proceed in a more just and equitable manner, taking into account demographic disparities in representation, as well as scalar challenges (the rural nature of work). The creation of a replicable rubric to guide workforce development assessments and potential collaborators gives other researchers an important tool to aid in future workforce planning.

However, several areas warrant further clarification or development—especially as it relates to the different dimensions they weight. I would like to see an acknowledgement of other dimensions of justice (not just procedural and representational). The authors may choose not to include this in their rubric, but they should make a case for why they have not chosen to include it. This can be achieved by bringing in more environmental justice-oriented scholarship.

Furthermore, the different weights/scores given in the rubric require greater justification—what is it about an HBCU versus an all-girls school that gives it different weight? Perhaps the point is self-evident, but given the gender disparity in the DW workforce, further elaboration could help to clarify the different weights given. Related to this, the paper presents the scoring metrics as "objective", but they are, indeed, normative assessments—I for one don't have an issue with this, but the authors may wish to consider more clearly stating that because the paper is focused on equity objectives anchored in different notions of justice, this influences their scoring criteria given the rubric's purpose (or something to that effect).

Overall, I do believe that the paper achieves its mission. It successfully introduces a rubric that can guide future workforce development efforts. Whether or not this rubric can be implemented right now (at least in the US), however, is another matter. I recommend accepting this paper with minor revisions to address the above critique, and to consider implementing a few other suggested changes to enhance the articles clarity and depth of the analysis.

**Specific suggestions:**

Line 29-31: I suggest that the authors revise the language given changes to the federal landscape.

Lines 134 to 143: The authors should explain why they did not work with workers, those actually employed by DW, and what they understand as the rationale for the workforce gap, including its demographic makeup.

Line 201: The authors should elaborate on why the radius selected is 100 miles. Why not 50 or some other number? They might also consider that 100 miles "as the crow flies"; i.e., in a straight line, is not how people travel in rural communities.

Line 213: The authors should explicitly reference to Schlosberg's work. A great deal has been written in response to his work, which should also be considered. They might also explain why distribution isn't considered as part of their criteria. This especially opens the rubric up to a critique of "tokenizing" or perpetuating a "checkbox" notion of justice.

Schlosberg, D. (2004). Reconceiving Environmental Justice: Global Movements And Political Theories. Environmental Politics, 13(3), 517–540. <a href="https://doi-org.ezproxy2.library.colostate.edu/10.1080/0964401042000229025">https://doi-org.ezproxy2.library.colostate.edu/10.1080/0964401042000229025</a>.

---

## Author Response (AR2)

**Building a Diverse and Equitable Distributed Wind Workforce: A Strategic Approach to Collaborator Selection**

WES-2024-145 | Author's Response Below are responses to referee report #2 in *blue italics*.

**Referee Comments**

This paper contributes to knowledge on how US workforce development efforts in distributed wind (DW) can proceed in a more just and equitable manner, taking into account demographic disparities in representation, as well as scalar challenges (the rural nature of work). The creation of a replicable rubric to guide workforce development assessments and potential collaborators gives other researchers an important tool to aid in future workforce planning.

However, several areas warrant further clarification or development—especially as it relates to the different dimensions they weight. I would like to see an acknowledgement of other dimensions of justice (not just procedural and representational). The authors may choose not to include this in their rubric, but they should make a case for why they have not chosen to include it.

Thank you for this thoughtful and reflective comment. Our engagement with the justice dimensions in this paper was not based on selective preference, but rather shaped by the scope, objectives, and methodological approach of the work. Specifically, our emphasis on demographic disparities and scalar challenges in distributed wind workforce development naturally aligned with procedural and recognition (representational) justice dimensions. These were the areas where our project activities could make direct and measurable contributions. Rather than prescribing in advance which justice dimensions to address, we allowed our project's goals and constraints to inform which dimensions were realistically achievable within this effort.

That said, we acknowledge the importance of other justice dimensions, particularly distributional justice, and have mentioned it on Line 237 in the revised manuscript to better situate our work within the broader environmental justice literature.

This can be achieved by bringing in more environmental justice-oriented scholarship.

We appreciate the recommendation to engage more environmental justice-oriented scholarship. While environmental justice literature, like foundational works by Schlosberg (2004), offers important early contributions, we view it as a conceptual precursor to more recent and field-relevant frameworks such as energy justice and energy equity, which we reference throughout the manuscript. These contemporary frameworks offer a more applicable lens for understanding justice in the context of distributed energy systems and workforce development, particularly because they have evolved to address the limitations of early liberal justice critiques, including those raised in Schlosberg's work. Further, the frameworks build on Schlosberg's key idea that justice is inherently multi-dimensional,

encompassing recognition, equity, and participation in outcomes. We have added text to this point starting on Line 237.

Furthermore, the different weights/scores given in the rubric require greater justification—what is it about an HBCU versus an all-girls school that gives it different weight? Perhaps the point is self-evident, but given the gender disparity in the DW workforce, further elaboration could help to clarify the different weights given.

We appreciate this comment and have expanded our justification in Section 2.2 - Rubric Development (Line 248) to clarify why certain institution types are assigned different point values. We explain that the weightings reflect both the scale of underrepresentation in the workforce with reference to relevant statistics (USEER 2024) on gender and race-based disparities in the sector.

Related to this, the paper presents the scoring metrics as "objective", but they are, indeed, normative assessments—I for one don't have an issue with this, but the authors may wish to consider more clearly stating that because the paper is focused on equity objectives anchored in different notions of justice, this influences their scoring criteria given the rubric's purpose (or something to that effect).

We fully agree. The revised manuscript now explicitly acknowledges that the rubric reflects normative choices based on equity objectives. This clarification is added in Section 2.2 (Line 212) and reinforced in Section 4 (Line 340), where we position the rubric as a strategic tool for advancing justice-driven priorities rather than a neutral selection mechanism.

Overall, I do believe that the paper achieves its mission. It successfully introduces a rubric that can guide future workforce development efforts. Whether or not this rubric can be implemented right now (at least in the US), however, is another matter. I recommend accepting this paper with minor revisions to address the above critique, and to consider implementing a few other suggested changes to enhance the articles clarity and depth of the analysis.

**Specific Suggestions**

Line 29-31: I suggest that the authors revise the language given changes to the federal landscape.

We chose to keep this language intact considering the federal landscape is still in flux as it relates to the Inflation Reduction Act (IRA) in particular, and the IRA itself was a key motivator for this work though it may be a historic document by the time this manuscript is published.

Lines 134 to 143: The authors should explain why they did not work with workers, those actually employed by DW, and what they understand as the rationale for the workforce gap, including its demographic makeup.

We added clarification that the first phase of this effort was focused on partner identification not engagement with the current workforce because of the focus on

foundation partnerships for hiring which was a previously identified immediate industry need (Line 135).

While we agree that a deeper exploration of the underlying causes of the workforce gap is important, doing so in detail was outside the scope of this phase of work. Our focus was on developing a collaborator identification framework grounded in equity metrics rather than diagnosing root causes. Subsequent phase of this work seeks to understand the rationale for distributed wind workforce gaps and this phased approach is already discussed briefly in Section 1 (line 77).

Line 201: The authors should elaborate on why the radius selected is 100 miles. Why not 50 or some other number? They might also consider that 100 miles "as the crow flies"; i.e., in a straight line, is not how people travel in rural communities.

We acknowledge the reviewer's important point that a 100-mile radius reflects straight-line ("as the crow flies") distance, which may not correspond directly to actual travel routes in rural areas. We selected a 100-mile radius to represent a regional catchment area around the focal point, based on both prior workforce development studies and practical considerations for postsecondary education access. The 100-mile radius is a commonly used planning metric that captures reasonable reach for rural populations while balancing data feasibility and geographic coverage. While this limitation is inherent in any radial analysis, we chose this method for its simplicity, replicability, and alignment with existing workforce and higher education planning frameworks. We now clarify this in the revised manuscript and note that travel times and road infrastructure are important considerations for future work (Line 207).

Line 213: The authors should explicitly reference to Schlosberg's work. A great deal has been written in response to his work, which should also be considered. They might also explain why distribution isn't considered as part of their criteria. This especially opens the rubric up to a critique of "tokenizing" or perpetuating a "checkbox" notion of justice.

Schlosberg, D. (2004). Reconceiving Environmental Justice: Global Movements And Political Theories. Environmental Politics, 13(3), 517–540. https://doi-org.ezproxy2.library.colostate.edu/10.1080/0964401042000229025.

We have responded to this comment above and the corresponding edits start on Lines 212, 248, and 340.